# Automaton Distillation: Neuro-Symbolic Transfer Learning for Deep Reinforcement Learning

**Precious Nwaorgu**[*]
*University of Central Florida*

*preciousmaduabuchi.nwaorgu@ucf.edu*

**Suraj Singireddy**[*,†]
*University of Texas at San Antonio*
*LinkedIn*

*ssingireddy@linkedin.com*

**Andre Beckus**
*Air Force Research Laboratory*

*andre.beckus@us.af.mil*

**Aden McKinney**
*Northeastern University*

*mckinney.ad@northeastern.edu*

**Mahyar Alinejad**
*University of Central Florida*

*mahyar.alinejad@ucf.edu*

**Chinwendu Enyioha**
*University of Central Florida*

*cenyioha@ucf.edu*

**Sumit Kumar Jha**
*University of Florida*

*sumit.jha@ufl.edu*

**Alvaro Velasquez**
*University of Colorado, Boulder*

*alvaro.velasquez@colorado.edu*

**George Atia**
*University of Central Florida*

*george.atia@ucf.edu*

**Reviewed on OpenReview:** *https://openreview.net/forum?id=Tyxmx2vNDb*

## Abstract

Reinforcement learning (RL) agents often struggle to reuse knowledge when task dynamics change, even when the underlying objective remains the same. This sample inefficiency is compounded by poor generalization beyond the training distribution. We introduce automaton distillation—a neuro-symbolic transfer learning approach that addresses both challenges by distilling Q-value estimates from a teacher agent into a compact automaton representation of the shared task objective. Critically, our method requires no explicit alignment between source and target state-action spaces: the automaton serves as a domain-agnostic intermediary through which value information is transferred. We propose two variants.

---

[*]Equal contributions.
[†]Work done while at the University of Texas at San Antonio.

Static transfer performs value iteration over the abstract MDP induced by the automaton, providing a lightweight initialization. Dynamic transfer distills empirical Q-values from a teacher's replay buffer onto automaton transitions, grounding symbolic abstractions in actual environment dynamics and correcting for mismatches between automaton trace length and true trajectory cost. We evaluate both variants on discrete and continuous gridworld tasks with sparse, non-Markovian rewards, and on a continuous benchmark. These results demonstrate that a shared symbolic objective is a sufficient bridge for effective few-shot transfer, even when source and target environments differ substantially in dynamics.

## 1 Introduction

Sequential decision-making tasks are foundational to autonomous systems, yet solving them remains a primary challenge for conventional reinforcement learning (RL). While deep RL has achieved remarkable success in stationary environments (Mnih & et al., 2015), its utility is often restricted by two critical bottlenecks: extreme sample inefficiency (Buckman et al., 2018) and poor adaptation when the task conditions change (Kirk et al., 2021). In most standard paradigms, a learned policy is inextricably tied to the specific state-action space of its training environment. Consequently, even a minor shift in task dynamics or layout typically necessitates retraining from scratch, as the agent fails to recognize that the underlying objective of the task remains invariant (Cobbe et al., 2019).

These limitations are most acute in tasks characterized by sparse or non-Markovian rewards (Kulkarni et al., 2016). In such scenarios, an agent must execute a precise, structured sequence of sub-goals before receiving any meaningful feedback, a setting where retraining from scratch is especially costly, since the agent must re-discover the correct subtask ordering entirely from experience. Existing approaches attempt to mitigate transfer cost by learning shared latent representations (Higgins et al., 2017) or constructing explicit inter-task mappings (Taylor & Stone, 2009). However, these methods typically require substantial additional data or manual engineering, and they assume some form of alignment between the source and target state-actions spaces.

In this work, we propose a different perspective: when source and target tasks share a common high-level objective, that objective is a sufficient bridge for knowledge transfer, without requiring any explicit alignment between between state-action spaces. Consider a robot trained to execute a multi-step delivery task in one building layout. When deployed in a new building, the physical observations differ entirely, yet the logical structure of the task: retrieve item, navigate to destination, return to base is identical. Standard transfer methods fail here because they attempt to map low-level states across environments. We instead transfer through the task structure itself.

We formalize this by expressing task objective as formal specification in finite-trace Linear Temporal Logic (LTL) (Pnueli, 1977; Camacho et al., 2017), which can be automatically compiled into Deterministic Finite Automata (DFAs) (Icarte et al., 2022). A DFA tracks symbolic task progress, which sub-goals have been completed, in what order independently of the specific observations in any particular environment. This makes it a natural domain-agnostic interface for knowledge transfer: the same automaton is valid in any environment that shares the objective, regardless of how that objective is physically realized.

We introduce automaton distillation, a transfer learning approach that exploits this structure. A teacher agent is trained to near-optimality in a source environment, and its learned Q-values are distilled onto the transitions of the shared task automaton. This produces a compact value summary that captures which symbolic transitions are rewarding, without encoding anything specific to the source environment's state space. A student in the target environment uses these transition-level estimates to bootstrap its own learning, receiving meaningful initial guidance without any direct correspondence between source and target observations.

We develop two variants of automaton distillation. Static transfer generates transition value estimates via value iteration over the abstract Markov Decision Process (MDP) induced by the automaton, a lightweight approach that requires a trained teacher, but does not account for actual environment dynamics. Dynamic transfer instead averages the teacher's Q-values over replay transitions that realize each automaton edge,

grounding symbolic value estimates in empirical environment dynamics. We show this distinction matters: when automaton trace length is a poor proxy for trajectory cost in the underlying environment, static transfer can produce negative transfer, while dynamic transfer corrects for this by drawing on teacher experience directly. This also connects to broader work on non-Markovian reward decision processes (Bacchus et al., 1996; Littman et al., 2017), where augmenting the state space with an automaton makes the problem Markovian in the product space (De Giacomo & Vardi, 2013; Gaon & Brafman, 2020), enabling standard RL algorithms to apply.

We evaluate automaton distillation on three gridworld environments with long horizons and sparse rewards, in both discrete and continuous variants, and on a continuous control task requiring sequential task completion. Transfer is tested across environments differing in map size and structure (discrete-to-discrete) and across fundamentally different state-action spaces (discrete-to-continuous).

The contributions of this paper are as follows:

- We introduce automaton distillation, a few-shot transfer learning framework for RL that requires no alignment between source and target state-action spaces, using a shared symbolic task objective as the sole transfer medium.

- We propose two variants *(i)* static transfer (value iteration over the automaton abstraction) and dynamic transfer (empirically grounded Q-value averaging over teacher replay) *(ii)* and characterize the conditions under which each is appropriate.

- We demonstrate empirically that automaton distillation improves transfer across discrete, continuous, and mixed-domain transfer settings.

The primary gap between this work and physical deployment is the labeling function: our experiments use handcrafted propositional functions over known state variables, whereas a real deployment would require atomic propositions derived from raw sensor observations.

## 2 Background

**Reinforcement Learning.** We model the RL environments using MDPs. An MDP is defined by a tuple $\mathcal{M} = \langle S, s_0, A, \mathcal{P}, R, \gamma \rangle$, where $S$ is the state space, $s_0 \in S$ is the initial state, $A$ is the action space, $\mathcal{P} : S \times A \times S \rightarrow [0, 1]$ is the transition kernel, $R : S \times A \times S \rightarrow \mathbb{R}$ is the reward function, and $\gamma \in [0, 1)$ is a discount factor. We denote the probability of transition from state $s$ to state $s'$ after taking action $a$ as $\mathcal{P}(s' \mid s, a)$. A policy $\pi$ is a map from states to probability distributions over actions. The value function $V^\pi(s) = \mathbb{E}_\pi \left[ \sum_{t=0}^\infty \gamma^t r_t \mid s_0 = s \right]$ is defined as the expected discounted return starting from the state $s$ and following $\pi$ the corresponding state-action value function is: $Q^\pi(s, a) = \mathbb{E}_\pi \left[ \sum_{t=0}^\infty \gamma^t r_t \mid s_0 = s, a_0 = a \right]$.

In many practical tasks, however, the reward signal depends not only on the current state but on the entire history of observations. Such tasks are more naturally modeled as Non-Markovian Reward Decision Processes (NMRDPs) (Bacchus et al., 1996), where $R : (S \times A)^* \rightarrow \mathbb{R}$ is defined over sequences of state-action pairs rather than individual transitions. Standard RL algorithms, which rely on the Markov assumption, cannot be applied directly. The key insight motivating our framework is that the non-Markovian structure of the reward signal can often be captured compactly using a formal task specification, which brings us to the role of automata.

**Finite-Trace Linear Temporal Logic.** We express task objectives using finite-trace Linear Temporal Logic (LTL$_f$) (Pnueli, 1977; De Giacomo & Vardi, 2015), a formalism for specifying structured sequences of events over finite trajectories. LTL$_f$ formulae are defined over a set of atomic propositions $AP$, which describes high-level features of the environment. Formulae are built from standard Boolean operators – negation ($\neg$), conjunction ($\wedge$), disjunction ($\vee$), and temporal operators – "until" ($\mathsf{U}$), "eventually" ($\mathsf{F}$), and "always" ($\mathsf{G}$).

Intuitively, $\mathsf{F}\varphi$ holds if $\varphi$ is is satisfied at some future timestep, $\mathsf{G}\varphi$ holds if $\varphi$ holds at every future timestep, and $\varphi_1 \mathsf{U} \varphi_2$ holds if $\varphi_1$ is satisfied at all timesteps before the first occurrence of $\varphi_2$. To connect the task

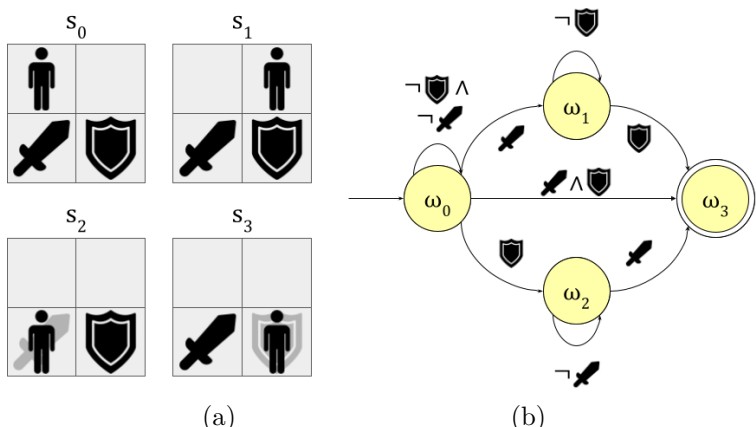

(a)          (b)

Figure 1: (a) At each time step, the agent may move one square in any cardinal direction. A sequence of actions satisfies the objective if and only if the agent obtains both the sword and the shield. The objective is decomposed using the atomic propositions $AP = \{\text{SWORD}, \text{SHIELD}\}$, with a labeling function $L$ such that $L(s_0) = \{\}, L(s_1) = \{\}, L(s_2) = \{\text{SWORD}\}, L(s_3) = \{\text{SHIELD}\}$. Rollouts which achieve the objective also satisfy the $\text{LTL}_f$ specification $\varphi = \mathbf{F}(\text{SWORD}) \wedge \mathbf{F}(\text{SHIELD})$. (b) An automaton defined over the alphabet $\Sigma = \{\{\}, \{\text{SWORD}\}, \{\text{SHIELD}\}, \{\text{SWORD, SHIELD}\}\}$. The automaton accepts the subset of strings in $\Sigma^*$ that satisfy the $\text{LTL}_f$ formula.

specification to the environment, we define a labeling function $L : S \to 2^{AP}$ that maps each state to the set of atomic propositions that hold in it. For example, in the environment shown in Figure 1a, $L$ maps grid positions to propositions such as SWORD or SHIELD depending on the objects present.

**Deterministic Finite-State Automaton (DFA).** To reason over the given the $\text{LTL}_f$ specification, we use a DFA, Therefore, given any LTL formulae $\varphi$, a corresponding DFA is defined by a tuple $\mathcal{A} = \langle \Sigma, \Omega, \omega_0, F, \delta \rangle$, where $\Sigma = 2^{AP}$ is the finite alphabet of the input language, $\Omega$ is the set of states with starting state $\omega_0$, $F \subseteq \Omega$ is the set of accepting states, and $\delta : \Omega \times \Sigma \to \Omega$ defines a state transition function. Tools for this compilation are readily available (Zhu et al., 2017), and during an episode, the automaton state is tracked in parallel with the environment: given the current automaton state $\omega$ and a new observation $s'$, the next automaton state is computed as $\omega' = \delta(\omega, L(s'))$.

Figure 1b depicts an example of a DFA for the formula $\varphi = \mathbf{F}(\text{SWORD}) \wedge \mathbf{F}(\text{SHIELD})$. The automaton starts in state $\omega_0$ and transitions to state $\omega_1$ or $\omega_2$ when it observes the propositions SWORD and SHIELD respectively, once both propositions have been observed (in either order), the automaton transitions to the accepting state $\omega_3$, indicating that the objective has been satisfied.

**Product MDP.** The product MDP $\mathcal{M}_\varphi$ of an MDP $\mathcal{M}$ and a DFA $\mathcal{A}$ synchronizes the environment state with the automaton state (Fu & Topcu, 2014). Its state space is the Cartesian product $S_\varphi = S \times \Omega$, and its transition kernel is given by

$$\mathcal{P}_\varphi((s', \omega') \mid (s, \omega), a) = \begin{cases} \mathcal{P}(s' \mid s, a), & \text{if } \omega' = \delta(\omega, L(s')), \\ 0, & \text{otherwise,} \end{cases}$$

where $\delta : \Omega \times \Sigma \to \Omega$ is the DFA transition function. The reward function $R' : \Omega \times \Sigma \to \mathbb{R}$ is a Markovian reward defined over automaton transitions. Because the automaton state encodes the task-relevant history, the product MDP satisfies the Markov property, and standard RL algorithms can be applied directly to the product state space $S \times \Omega$. In practice, one does not need to construct the Cartesian product explicitly; instead, the automaton state is maintained alongside the environment state and concatenated with the environment state as input to the policy or value network.

**Problem Setting: Transfer via Shared Objectives.** The framework above establishes how a single agent solves a structured task in one environment. We consider two agents, a teacher operating in a source

environment $\mathcal{M}_\tau$ and a student learning in a target environment $\mathcal{M}_\kappa$, that share a common high-level objective $\varphi$ but may differ in every other respect. Formally, we make no assumption that $S_\tau = S_\kappa$ or $A_\tau = A_\kappa$: the state spaces, action spaces, and transition dynamics may all differ. The two environments share only three things: a common set of atomic propositions $AP$, a labeling function $L$ that maps states in each environment to truth assignments over $AP$ (with potentially different implementations, e.g., grid adjacency versus Euclidean distance thresholds), and the task objective $\varphi$ and its compiled DFA $\mathcal{A}$. Because $\mathcal{A}$ is defined entirely over $AP$ and not over the raw observations of either environment, it acts as a domain-agnostic representation of task progress that is valid in both settings. The central question is: can the teacher's learned Q-values be used to accelerate student learning, given that there is no direct correspondence between their state-action spaces? We answer affirmatively; by associating value estimates with automaton transitions rather than raw state-action pairs, we transfer knowledge through the shared symbolic structure of the task itself. This is the key idea behind automaton distillation, which we develop in Section 3.

## 3 Automaton Distillation

We now describe our approach, the core idea is straightforward: rather than transferring knowledge directly from source to target state-action pairs, which requires alignment between the two spaces, we use the shared task automaton as an intermediate representation. The teacher's learned Q-values are compressed onto automaton transitions, producing a compact value summary that is environment-agnostic. The student then uses this summary to initialize its learning in the target environment. The method proceeds in three stages, illustrated in Figure 2, a DQN (or TD3) agent is trained to near-optimality in the source environment. Its experience replay buffer is populated with augmented transitions that include the automaton state. In section Section 3.2, we describe the distillation process where the Q-value estimates are associated with each transition in the shared task automaton, either via value iteration (static transfer) or by averaging teacher Q-values over replay transitions that realize each automaton edge (dynamic transfer). In Section 3.3, we describe how the student agent uses these transition values to modify its Q-learning targets early in training, with the teacher's influence annealed to zero as the student accumulates its own experience.

### 3.1 Teacher Training

The teacher agent is trained using standard deep RL methods in the source environment $\mathcal{M}_\tau$. Specifically, we use a Dueling DQN (Wang et al., 2016b) for discrete environments and TD3 (Fujimoto et al., 2018) for continuous environments, augmented to operate over the product MDP state space $S_\tau \times \Omega$. At each timestep, the teacher observes state $s$, selects action $a$, receives reward $r$, and transitions to $s'$. The new automaton state is computed as $\omega' = \delta(\omega, L(s'))$, and the full augmented transition $((s, \omega), a, r, (s', \omega'))$ is stored in a replay buffer ER. We define $\eta_\tau(\omega, \sigma)$ as the number of times automaton node $\omega$ and proposition set $\sigma \in 2^{AP}$ co-occur in ER, that is, the number of replay transitions that realize the automaton edge $\omega \xrightarrow{\sigma} \omega'$ :

$$\eta_\tau(\omega, \sigma) = |\{((s, \omega), a, r, (s', \omega')) \in ER | L(s') = \sigma\}|. \tag{1}$$

This count is used to give the average view of the source environment and how it would relate to the target.

### 3.2 Distillation: Associating Values with Automaton Transitions

Once the teacher is trained, we associate a scalar Q-value estimate with each transition $(\omega, \sigma)$ in the automaton. This estimate captures how desirable it is to realize that symbolic transition, that is, to progress from automaton state $\omega$ by observing proposition $\sigma$. We propose two methods for generating these estimates.

#### 3.2.1 Static Automaton Transfer

Static transfer generates transition value estimates without using the teacher's learned Q-function. Instead, value iteration is performed directly over the abstract MDP induced by the automaton, treating automaton nodes as states and transitions as actions:

$$Q(\omega, \sigma) \leftarrow Q(\omega, \sigma) + \alpha(R'(\omega, \sigma) + \gamma \max_{\sigma'} Q(\omega', \sigma') - Q(\omega, \sigma)), \tag{2}$$

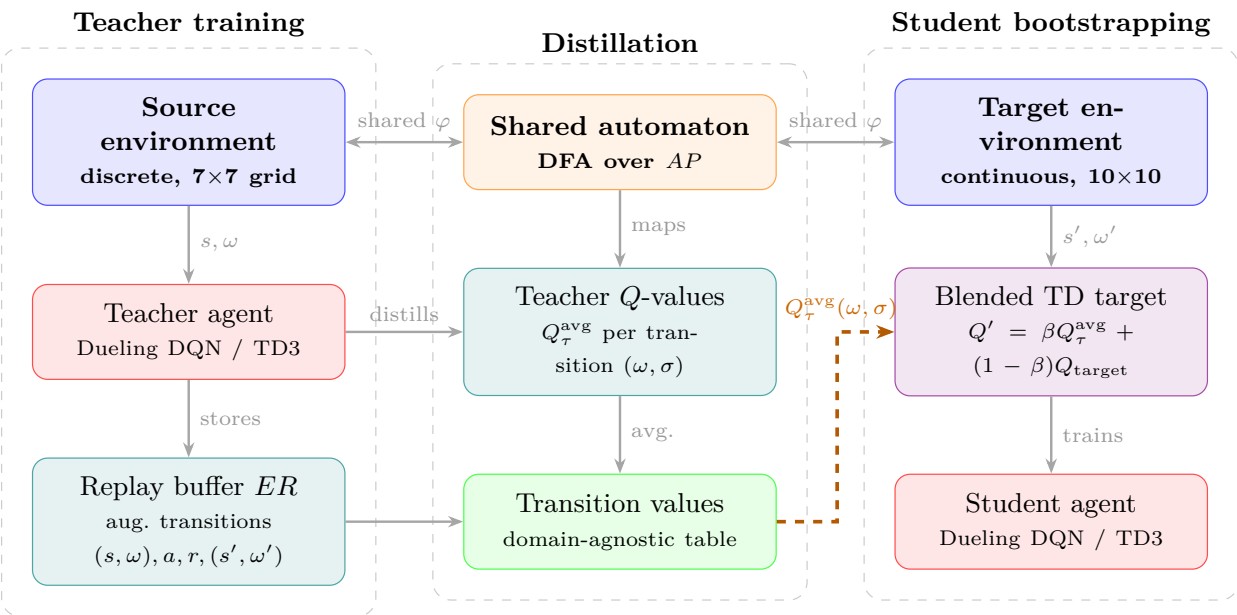

Figure 2: **Automaton distillation pipeline.** *Left:* The teacher agent is trained in the source environment over the product MDP state space; augmented transitions are stored in replay buffer *ER*. *Middle:* Teacher $Q$-values are averaged over replay transitions that realize each automaton edge $\omega \xrightarrow{\sigma} \omega'$, producing a table of scalar transition values $Q_\tau^{\mathrm{avg}}(\omega, \sigma)$. *Right:* The student uses these values to bias its TD targets early in training via annealing coefficient $\beta$. The dashed arrow marks the *sole artifact transferred*, no raw states, actions, or network weights cross the teacher–student boundary. Both agents compile the same DFA from the shared $\mathrm{LTL}_f$ specification independently.

where $R'(\omega, \sigma)$ is the reward associated with transitioning from $\omega$ under proposition $\sigma$, and $\omega' = \delta(\omega, L(s'))$. Static transfer is lightweight and requires no trained teacher. However, it has a fundamental limitation: value iteration over the abstract automaton optimizes for short traces under discounting, without any knowledge of how costly each transition is to realize in the actual environment. When trace length in the automaton is a poor proxy for trajectory length in the environment—for instance, when a symbolically short path requires many low-level steps—static transfer can assign misleading values that actively slow student learning. This becomes a problem when the symbolic structure of the automaton does not reflect the physical structure of the task.

Consider the objective $\varphi = \mathsf{F}(b \lor e) \land (\neg\mathsf{F}(a) \lor \neg\mathsf{F}(c)) \land (a \mathbin{\mathsf{R}} \neg b) \land (c \mathbin{\mathsf{R}} \neg d) \land (d \mathbin{\mathsf{R}} \neg e)$, which compiles into the two-trace automaton shown in Figure 3a. The two completion paths are $a \to b$ (2 transitions) and $c \to d \to e$ (3 transitions). Under discounting, static transfer assigns higher value to $a \to b$ because it appears shorter in the automaton. However, in the actual environment the path $c \to d \to e$ may be physically shorter; for example, if the objects corresponding to $a$ and $b$ are far apart but $c, d$ and $e$ are clustered nearby. Static transfer has no way to detect this mismatch; it sees only the automaton structure, not the environment dynamics. This failure is not pathological. It arises naturally whenever objects or waypoints are unevenly distributed in the environment, obstacles alter navigation costs, or actions have variable execution time. This limitation motivates our second variant, which replaces symbolic reasoning with empirical grounding.

### 3.3 Dynamic Automaton Transfer

Dynamic transfer addresses the limitation above by grounding transition value estimates in the teacher's empirical experience. For each automaton transition $(\omega, \sigma)$, we compute the average Q-value assigned by

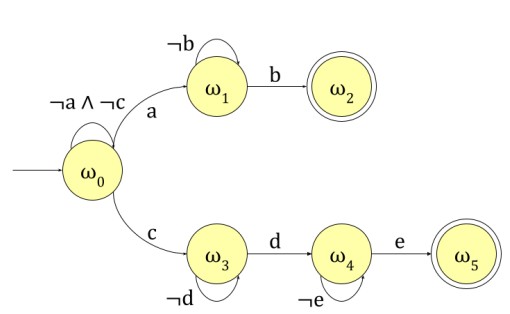

(a) Simple automaton with two traces.

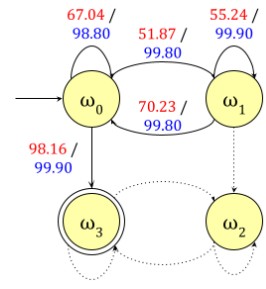

(b) Teacher Q-values produced by dynamic (red) and static (blue) automaton distillation on the *Blind Craftsman* environment.

Figure 3: (a) A two-trace automaton compiled from the objective $\varphi = \mathsf{F}(b \vee e) \wedge (\neg\mathsf{F}(a) \vee \neg\mathsf{F}(c)) \wedge (a \, \mathsf{R} \, \neg b) \wedge (c \, \mathsf{R} \, \neg d) \wedge (d \, \mathsf{R} \, \neg e)$. (b) Teacher Q-values produced by dynamic (red) and static (blue) automaton distillation on the Blind Craftsman environment. Transitions which were not observed in the environment are denoted by dotted lines.

the teacher's network across all replay transitions that realize that edge:

$$Q_\tau^{\mathrm{avg}}(\omega, \sigma) = \frac{\sum_{\{((s,\omega),a,r,(s',\omega')) \in ER | L(s') = \sigma\}} Q_\tau((s,\omega),a)}{\eta_\tau(\omega,\sigma)}. \tag{3}$$

where $Q_\tau((s,\omega),a)$ is the Q-value assigned by the teacher's network to the augmented state-action pair $((s,\omega),a)$, and $\eta_\tau(\omega,\sigma)$ is defined in equation 1. Intuitively, $Q_\tau^{\mathrm{avg}}(\omega,\sigma)$ captures the empirical desirability of realizing transition $(\omega,\sigma)$ as experienced by the teacher. Crucially, this estimate implicitly reflects the true cost of realizing each symbolic transition in the environment, including how many low-level steps it requires and how easily it can be achieved, information that the abstract automaton alone cannot represent. This makes dynamic transfer robust to cases where the automaton structure misrepresents task difficulty.

### 3.3.1 Student Bootstrapping

Given the automaton transition values $Q_\tau^{\mathrm{avg}}(\omega, \sigma)$ (from either static or dynamic transfer), we use them to bias the student's Q-learning targets early in training. At each training step, the standard Bellman target for the student is blended with the teacher's automaton transition value:

$$Q'_\kappa((s,\omega),a) = \beta(\omega, L(s')) \, Q_\tau^{\mathrm{avg}}(\omega, L(s')) + \big(1 - \beta(\omega, L(s'))\big) Q_{target}. \tag{4}$$

where $Q_\tau^{\mathrm{avg}}(\omega, L(s'))$ is the teacher's value for the current automaton transition, $Q_{target}$ is the standard Bellman target, and $\beta(\omega, \sigma) \in [0, 1]$ controls the relative influence of the two. The standard target $Q_{target}$ is defined as:

$$Q_{\mathrm{target}} = r + \gamma \max_{a'} Q_{\mathrm{student}}((s',\omega'),a';\theta^{\mathrm{target}}), \tag{5}$$

where $\theta^{\mathrm{target}}$ denotes the parameters of the target Q-network. For TD3, the corresponding target is

$$Q_{\mathrm{target}} = r + \gamma \min_{i=1,2} Q_{\mathrm{student}}\big((s',\omega'),\tilde{a};\theta_i^{\mathrm{target}}\big). \tag{6}$$

where $\theta_i^{\mathrm{target}}$ denotes the parameters of the $i$-th target critic, and $\tilde{a} = \pi_{\phi'}(s') + \varepsilon$ is the target policy's action with exploration noise $\varepsilon$.

The annealing function is defined as $\beta(\omega, \sigma) = \rho^{\eta_{\mathrm{student}}(\omega,\sigma)}$, where $\rho = 0.999$ and $\eta_{\mathrm{student}}(\omega, \sigma)$ is the number of times the transition $(\omega, \sigma)$ has been sampled during student training. As the student accumulates experience on a given automaton transition, $\beta$ decays exponentially toward zero, and the teacher's influence on that transition is phased out. When $\beta = 0$ for all transitions, the student's update reduces exactly to standard DQN or TD3. This per-transition annealing is important: it allows the teacher to provide strong

guidance on transitions the student has rarely seen, while stepping back on transitions the student has already learned well. The full student-learning procedure, including the target-network updates, is given in Algorithm 1 in Appendix A.

The student's network is updated by minimizing the prioritized TD error against the modified target:

$$\text{Loss}(\theta) = \mathbb{E}_{((s,\omega),a,r,(s',\omega'))\sim P(ER)}[Q'_{\text{student}}((s,\omega),a) - Q((s,\omega),a;\theta)]^2, \tag{7}$$

where $P$ is a priority function that upweights transitions with higher prediction error (Schaul et al., 2015)

The asymptotic behavior of automaton Q-learning depends on the annealing function $\beta$ in the student Q-udpate in equation 4. When $\beta = 0$, automaton Q-learning reduces to vanilla Q-Learning, confirming that the teacher's influence is fully phased out as the student gains experience. To provide theoretical grounding for this blending mechanism, specifically, to show that the per-transition annealing coefficient produces convergent Q-value estimates rather than destabilizing training, we establish a convergence result for the tabular special case. We note that this result applies to finite state and action spaces and does not directly extend to the function approximation setting used in practice; Remark 1 below discusses the implications for DQN and TD3.

**Theorem 1** (Convergence of Tabular Automaton Q-Learning). *Consider the product MDP $\mathcal{M}_\varphi = \langle S \times \Omega, (s_0, \omega_0), A, T \times \delta, R' \rangle$, where $R' : \Omega \times \Sigma \to \mathbb{R}$ is the reward function defined over automaton transitions. Define $\sigma_t = L(s_{t+1})$ and $r_t := R'(\omega_t, \sigma_t)$ as the reward observed during step $t$, corresponding to the transition $\omega_t \xrightarrow{\sigma_t} \omega_{t+1} = \delta(\omega_t, \sigma_t)$. Suppose the tabular Q-values are updated according to:*

$$Q_{t+1}((s_t, \omega_t), a_t) = (1 - \alpha_t)Q_t((s_t, \omega_t), a_t) + \alpha_t \beta_t Q^{avg}_{teacher}(\omega_t, \sigma_t)$$
$$+ \alpha_t(1 - \beta_t)\left[r_t + \gamma V_t(s_{t+1}, \omega_{t+1})\right], \tag{8}$$

*where $V_t(s, \omega) = \max_a Q_t((s, \omega), a)$. Then, $Q_t((s, \omega), a) \to Q^*((s, \omega), a)$ with probability 1 under the following conditions:*

1. *The state space $S$, automaton state space $\Omega$, and action space $A$ are finite.*

2. *The learning rates $\alpha_t \in [0, 1)$ satisfy $\sum_t \alpha_t = \infty$ and $\sum_t \alpha_t^2 < \infty$.*

3. *The distillation weights $\beta_t \geq 0$ satisfy $\lim_{t\to\infty} \beta_t = 0$ and $\sum_t \alpha_t(1 - \beta_t) = \infty$.*

4. *The symbolic reward variance $\text{Var}(r_t)$ is uniformly bounded.*

5. *Either $\gamma = 1$ and all policies reach a cost-free terminal state, or $\gamma \in [0, 1)$.*

*Proof.* We decompose the Q-values as $Q_t = q_t + h_t$ with updates:

$$q_{t+1}((s_t, \omega_t), a_t) = (1 - \alpha_t)q_t((s_t, \omega_t), a_t) + \alpha_t(1 - \beta_t)\left[r_t + \gamma V_t(s_{t+1}, \omega_{t+1})\right], \tag{9}$$
$$h_{t+1}((s_t, \omega_t), a_t) = (1 - \alpha_t)h_t((s_t, \omega_t), a_t) + \alpha_t \beta_t Q^{\text{avg}}_{\text{teacher}}(\omega_t, \sigma_t). \tag{10}$$

*Convergence of $q_t$:* This update corresponds to Q-learning in the product MDP using rewards $r_t = R'(\omega_t, \sigma_t)$. Since the reward is bounded, the learning rate $\alpha_t(1 - \beta_t)$ satisfies Robbins–Monro conditions, and the state-action space is finite, it follows that (e.g., (Jaakkola et al., 1993)):

$$q_t((s, \omega), a) \to Q^*((s, \omega), a) \quad \text{w.p. 1.}$$

*Convergence of $h_t$:* Let $c_t := Q^{\text{avg}}_{\text{teacher}}(\omega_t, \sigma_t)$. Since $c_t$ is drawn from a fixed and bounded function over the finite domain $\Omega \times \Sigma$, there exists $C < \infty$ such that $|c_t| \leq C$ for all $t$. The update becomes:

$$h_{t+1} = (1 - \alpha_t)h_t + \alpha_t \beta_t c_t.$$

This is a stochastic approximation with a vanishing forcing term $\beta_t c_t$, and under the given assumptions on $\alpha_t$ and $\beta_t$, it converges to zero almost surely.

Since $q_t \to Q^*$ and $h_t \to 0$ almost surely, we conclude:

$$Q_t = q_t + h_t \to Q^* \quad \text{w.p. 1.} \quad \blacksquare$$

$\square$

**Remark 1.** *Theorem 1 applies to the tabular setting. The practical algorithm uses function approximation (DQN or TD3), where standard convergence guarantees do not hold. However, the annealing condition $\beta_t \to 0$ ensures that the teacher's influence vanishes as training proceeds, and the student's update asymptotically reduces to standard DQN or TD3. This is consistent with the empirical stability observed in our experiments.*

## 4  Related Work

The challenge of reusing knowledge between environments with different state spaces, action spaces, or dynamics has been studied under several paradigms (Zhu et al., 2023). Inter-task mapping approaches construct explicit correspondences between source and target state-action spaces, either manually (Taylor & Stone, 2005)or via learned latent subspaces (Ammar & Taylor, 2012; Ammar et al., 2015), but require simultaneous access to both environments and become infeasible when spaces differ structurally. Representation transfer methods, including DARLA (Higgins et al., 2017), CURL (Srinivas et al., 2020), and Latent Unified State Representation (Xing et al., 2021), learn domain-invariant features but require alignment between observation spaces and are not designed for tasks with non-Markovian reward signals.

Meta-learning approaches such as MAML (Finn et al., 2017) and related approaches (Wang et al., 2016a) learn fast-adapting initializations but require a distribution of training tasks rather than a single source environment. None of these methods transfer without some form of state-space alignment, and none exploit a shared formal task specification as the transfer medium.

The reward machine (RM) framework (Icarte et al., 2018; 2022) is the standard formalism for non-Markovian reward signals in RL, and our method builds directly on it. However, existing RM-based methods differ from ours in important ways. CRM (Icarte et al., 2022) improves data efficiency by relabeling transitions with counterfactual RM states, generating additional training signal within a single environment. It does not transfer knowledge from a teacher and provides no mechanism to bootstrap learning in a structurally different target domain. Automaton-guided reward shaping (Camacho et al., 2017; 2018; 2019) and CPREP (Azran et al., 2024) incorporate prior knowledge by modifying the reward signal. Both assume the source and target environments share sufficient dynamics for the shaped reward to remain valid, an assumption that breaks down in cross-domain settings, as our Gold Mine and discrete-to-continuous results demonstrate.

The critical technical distinction is where prior knowledge enters the learning process. Reward shaping and CPREP modify the reward term permanently; our method modifies the TD target directly in the loss function via a per-transition annealing coefficient $\beta$ in Equation (4). As the student accumulates experience on each automaton transition, $\beta$ decays toward zero and the update reduces to standard DQN or TD3. This allows the student to override poor teacher estimates without residual interference, a property reward shaping methods do not possess.

## 5  Experimental Evaluation

We evaluate automaton distillation across two categories of environments: structured gridworld tasks with discrete and continuous variants, and continuous control benchmarks. Our evaluation is designed to answer three questions: *(i)* Does automaton distillation reduce the steps required to reach near-optimal performance compared to reward-machine transfer baselines?*(ii)* Does dynamic transfer outperform static transfer, particularly when automaton trace length misrepresents actual task difficulty?*(iii)* Does the method remain effective when the teacher and student operate in structurally different environments, including different state-action spaces and different physical dynamics?

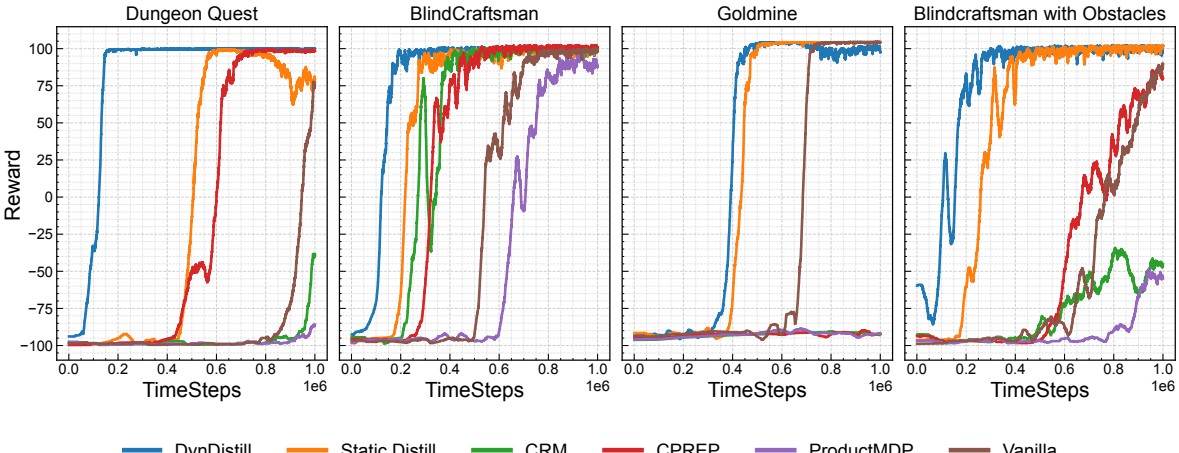

Figure 4: Reward per episode (y-axis) over time (x-axis) during training in the discrete student environment.

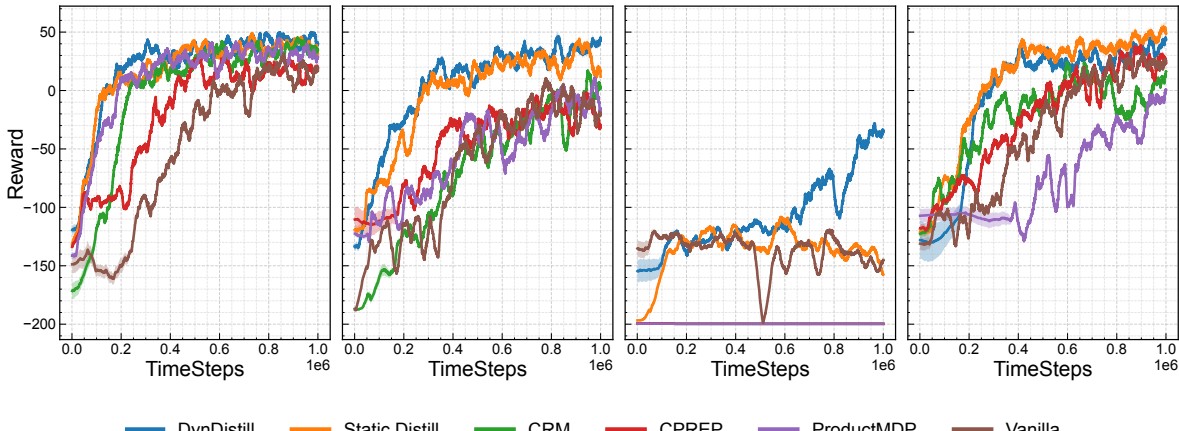

Figure 5: Reward per episode (y-axis) over time (x-axis) during training in the continuous student environment.

## 5.1 Experimental Setup

**Environments.** We evaluate across four environment families of increasing complexity, covering both discrete and continuous state-action spaces. For a discrete gridworld, the source environment is a randomly generated $7 \times 7$ grid and the target is an independently generated $10 \times 10$ grid with atomic proposition locations randomly sampled in each. The student therefore faces a different map layout from the teacher, with no overlap in the positions of task-relevant objects. We additionally evaluate a discrete-to-continuous variant in which the teacher is trained in the discrete $7 \times 7$ gridworld and the student operates in a physically simulated continuous environment of equivalent scale, powered by a Box2D engine. The student's state space consists of continuous 2D position and velocity, and objects are collected by proximity (Euclidean distance $< 0.25$m) rather than grid occupancy. This setting tests transfer across fundamentally different state-action spaces and physical dynamics. Also, we evaluate on the continuous FlatWorld environment (Voloshin et al., 2023) in which multiple atomic propositions can hold simultaneously. This tests automaton distillation under overlapping proposition semantics, a setting not covered by the gridworld experiments. Further implementation details for all environments are provided in Appendix B.1

**Teacher-Student Transfer Setup.** In all experiments, the teacher is trained to near-optimality in the source environment using standard DQN (Wang et al., 2016b)/TD3 (Fujimoto et al., 2018). Its replay buffer is then used to compute automaton transition values $Q_\tau^{\mathrm{avg}}(\omega, \sigma)$ via equation 3, which are transferred to the student as described in Section 3. The student is initialized from scratch in the target environment with no access to the teacher's environment or network weights, only the distilled automaton transition values are transferred. Task objectives are specified as $\mathrm{LTL}_f$ formulae over the atomic propositions $AP$ for each environment and compiled into DFAs using the FLLOAT synthesis tool. For the gridworld tasks, the $APs$

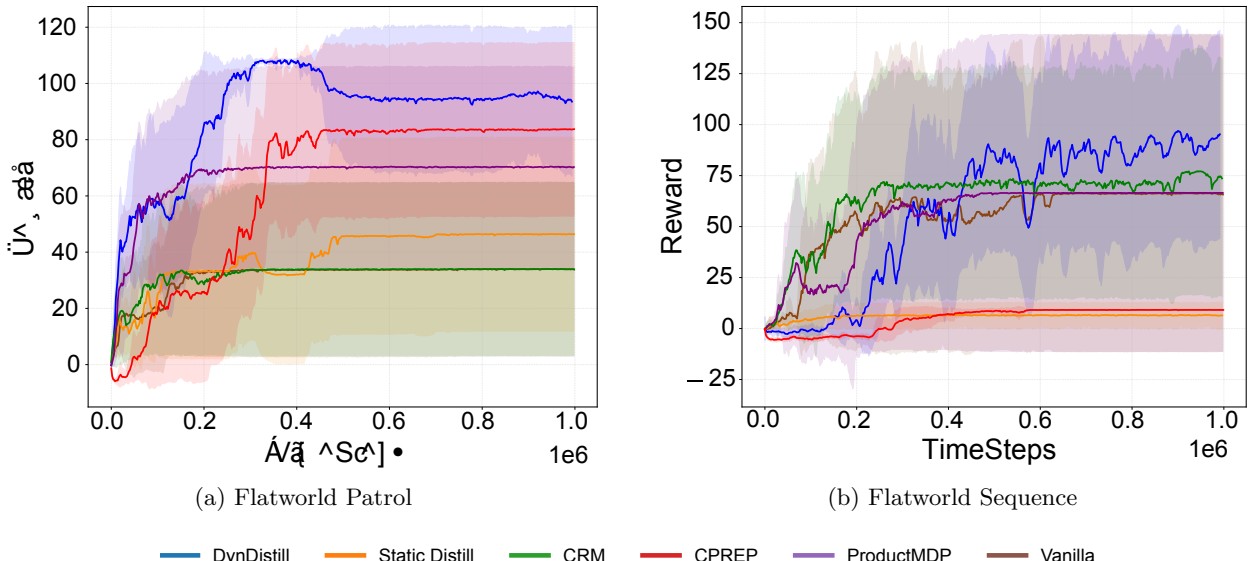

(a) Flatworld Patrol           (b) Flatworld Sequence

Figure 6: Reward per episode (y-axis) over timesteps (x-axis) during training in (a) Flatworld Patrol and (b) Flatworld Sequence Environments.

correspond to task-relevant objects (e.g., wood, factory, home in Blind Craftsman; key, sword, shield, dragon in Dungeon Quest). For FlatWorld, the *AP*s correspond to zone membership, evaluated at each timestep via the labelling function $L$. Complete task specifications and automata for each environment are provided in Appendix B.2.

**Baselines.** To fairly assess the contribution of our approach, we primarily compare against methods that also exploit automaton or reward-machine structure, so that performance gaps among these methods reflect differences in how symbolic task knowledge is used. We also include Vanilla DQN/TD3 as a non-symbolic reference baseline. Our baselines fall into two categories. *Single-environment* baselines operate entirely within the student environment and use no knowledge from a teacher agent: CRM (Icarte et al., 2022) generates counterfactual experiences by relabeling transitions with alternative automaton states; Product MDP augments the agent's state with the current automaton node but adds no transfer; and Vanilla DQN/TD3 uses no automaton augmentation or transfer. *Transfer baselines* explicitly move knowledge from a source environment to the student: CPREP (Azran et al., 2024) pre-plans over the reward machine abstraction and incorporates the resulting guidance as a reward bonus in the target environment. Static and dynamic automaton distillation are reported separately. Outperforming CRM is a meaningful benchmark, as it indicates that the transferred knowledge is genuinely additive beyond what the student can learn on its own in the target environment.

**Evaluation Metrics.** We report two transfer metrics following (Taylor & Stone, 2009): the transfer ratio (TR), defined as the ratio of the cumulative reward obtained by the transfer agent to that obtained by an agent trained from scratch, where TR > 1 indicates positive transfer; and time-to-threshold AUC ($TT_{AUC}$), defined as the area under the curve of time-to-threshold values computed over a range of performance thresholds, which provides a comprehensive measure of convergence speed across the training curve. All results are averaged over 8 random seeds and reported as mean and standard deviation. We note that the Vanilla and Product MDP baselines learn entirely in the student environment and do not involve transfer from a teacher agent. Therefore, we do not evaluate transfer metrics for these baselines; instead, we report their reward curves over training iterations. Unless indicated otherwise, the methods are trained for 1M interaction steps on each environment. Full hyperparameter settings and network architectures are provided in Appendix B.3.

Table 1: An evaluation of automaton distillation and other reward machine transfer approaches. We show that automaton distillation successfully transfers knowledge to the student environment. We show the mean and standard deviation of the transfer ratio averaged over 8 seeds. The italicized figures indicate negative TR. *BlindCraftsManObs* refers to the environment Blindcraftsman with obstacles.

| Metric | Transfer-Dynamics | Environment | CRM | CPREP | Static Distill | Dyn Distill |
|---|---|---|---|---|---|---|
| $\text{TT}_{\text{AUC}}$ | Discrete–Discrete | BlindCraftsMan | 199835.98 ± 39626.54 | 237009.07 ± 30064.85 | 194037.82 ± 9396.00 | **111515.92 ± 2818.89** |
| | | BlindCraftsManObs | 434982.79 ± 122704.47 | 392687.42 ± 99958.54 | 186997.95 ± 23397.48 | **38668.11 ± 9272.46** |
| | | DungeonQuest | 827653.48 ± 119397.18 | 438406.01 ± 44567.97 | 375656.36 ± 61651.82 | **90682.37 ± 15536.00** |
| | | GoldMine | 1000000.0 ± 0.0 | 1000000.0 ± 0.0 | **290313.72 ± 43304.58** | 723871.45 ± 296460.19 |
| | Discrete–Continuous | BlindCraftsMan | 335160.81 ± 42506.67 | 311190.14 ± 78927.61 | 175176.02 ± 66449.40 | **126996.97 ± 45392.95** |
| | | BlindCraftsManObs | 437139.52 ± 53840.96 | 299458.60 ± 63050.75 | **154929.25 ± 53437.07** | 216010.25 ± 51547.48 |
| | | DungeonQuest | 213265.48 ± 41918.38 | 283279.45 ± 35904.54 | 149336.94 ± 71913.11 | **196528.84 ± 54569.72** |
| | | GoldMine | 1000000.0 ± 0.0 | 1000000.0 ± 0.0 | 588773.29 ± 123336.65 | **469081.81 ± 71926.05** |
| TR | Discrete–Discrete | BlindCraftsMan | 10.17 ± 0.017 | 10.93 ± 0.02 | 9.83 ± 0.03 | **12.50 ± 0.02** |
| | | BlindCraftsManObs | *−1.06 ± 0.03* | 0.24 ± 0.2 | 2.45 ± 0.02 | **3.013 ± 0.03** |
| | | DungeonQuest | *−3.54 ± 0.11* | 19.87 ± 0.02 | 20.58 ± 0.10 | **43.54 ± 0.39** |
| | | GoldMine | *−3.86 ± 0.15* | *−2.76 ± 0.10* | **0.90 ± 0.16** | 0.86 ± 0.16 |
| | Discrete–Continuous | BlindCraftsMan | *−0.05 ± 0.048* | 0.29 ± 0.062 | 0.94 ± 0.022 | **1.073 ± 0.05** |
| | | BlindCraftsManObs | 0.43 ± 0.058 | 0.37 ± 0.06 | **1.22 ± 0.030** | 0.882 ± 0.067 |
| | | DungeonQuest | 0.88 ± 0.028 | 1.07 ± 0.045 | 1.28 ± 0.05 | **1.36 ± 0.056** |
| | | GoldMine | *−0.45 ± 0.001* | *−0.45 ± 0.001* | 0.009 ± 0.020 | **0.20 ± 0.046** |

## 5.2 Results

**Discrete-Discrete Transfer:** Dynamic distillation achieves the fastest convergence and highest TR across all four environments (Figure 4, Table 1), with the largest gains in Dungeon Quest (TR = 43.54 vs. 19.87 for CPREP). Static distillation also transfers positively in this setting, confirming that the automaton abstraction is a reliable guide when source and target environments share discrete dynamics. The exception is Gold Mine, where static distillation produces only minimal transfer advantage (TR = 0.90) : value iteration over the automaton favors the diamond route, which has fewer transitions but requires collecting 30 iron, over the empirically faster gold mine route. Dynamic distillation avoids this by grounding transition values in teacher replay, correctly reflecting actual trajectory costs. CRM and CPREP also exhibit negative TR in Gold Mine (−3.86 and −2.76 respectively), suggesting that purely symbolic value estimates are unreliable when automaton structure misrepresents task difficulty.

**Gridworld Discrete-Continuous Transfer:** Dynamic distillation outperforms all baselines across all environments (Figure 5, Table 1). Static distillation, which performed competitively in the discrete-to-discrete setting, degrades here; the change in dynamics makes its value estimates less reliable, and negative transfer emerges in several environments. This contrast directly illustrates the core limitation of static transfer: it relies on the automaton abstraction remaining a reasonable proxy for the target environment, which becomes less valid as the dynamics gap grows. Dynamic distillation maintains consistent positive transfer across all environments, including settings where CRM produces negative TR (BlindCraftsman: −0.05, Gold Mine: −0.45), confirming that empirical grounding is essential for robust cross-domain transfer.

**FlatWorld.** Figures 6a and 6b report results on the FlatWorld Patrol and Sequence tasks, in which multiple atomic propositions can hold simultaneously due to overlapping circular regions. Here, the teacher is trained in an open layout and the student must navigate around walls and obstacles that did not exist in the teacher's environment. Dynamic distillation outperforms all baselines considerably. This setting is notable because overlapping propositions mean the labeling function $L$ can fire multiple APs at the same timestep, creating ambiguity in automaton transition attribution. Despite this, dynamic distillation's empirical averaging over replay transitions handles simultaneous proposition activations naturally, without any modification to the distillation procedure. Static distillation performs competitively in the Patrol task but degrades in the Sequence task, where the three-step ordering constraint makes symbolic trace length a less reliable proxy for actual task difficulty.

These findings confirm that our approach enhances transfer performance even when environmental dynamics differ significantly. Dynamic automaton distillation utilizes an empirical estimate of trajectory length over the teacher's decision process and circumvents inaccuracies in the abstract MDP. Thus, dynamic automaton distillation is effective when the optimal policies in the teacher and student environments follow similar automaton traces.

Some behavioral specifications can lead to objective automata with cycles, as evidenced by the *Blind Crafts-man* environment. Unlike static distillation, dynamic distillation can distinguish between states which are (nearly) equivalent in the abstract MDP by incorporating episode length; states which are further from the goal receive discounted rewards, resulting in smaller Q-values as shown in Figure 3b

Cycles in the automaton do not necessarily lead to infinite reward loops as it is often the case that in the original environment, the cycle may be taken only a finite number of times. While it is possible to construct an automaton without cycles by expanding the state space of the automaton to include the number of cycles taken, the maximum number of cycle traversals must be known *a priori* and incorporated into the objective specification, which may not be possible. Additionally, environments that share an objective may admit different numbers of cycle traversals; thus, cycles offer a compact representation that permits knowledge transfer between environments. However, cycles can aggravate the differences between the abstract MDP and the original decision process, resulting in negative knowledge transfer. In such cases, state-of-the-art transfer methods (Icarte et al., 2022) may actually *increase* training time relative to a naïve learning algorithm.

# 6    Conclusion

We introduced automaton distillation, a few-shot transfer learning framework that requires no alignment between source and target state-action spaces. A teacher's Q-values are compressed onto a shared task automaton, producing an environment-agnostic value summary that bootstraps student learning across structurally different environments.

Static transfer assigns transition values through value iteration over the automaton abstraction, making it lightweight but sensitive to cases in which trace length misrepresents task difficulty. Dynamic transfer grounds these estimates in teacher replay, correcting for such misrepresentation. This distinction is consequential: dynamic distillation maintains positive transfer where static transfer fails, and consistently has good performance, confirming that transferred automaton knowledge is genuinely additive beyond efficient in-environment learning.

Results across discrete, continuous, and mixed-domain settings demonstrate that a shared formal task objective is a sufficient bridge for effective transfer. Extending this to settings where propositions must be inferred from raw observations is the primary open challenge and a direction for future work.

# 7    AI Impact Statement

Automaton distillation reduces the cost of retraining RL agents when task dynamics shift, by transferring knowledge through a shared symbolic objective rather than raw state-action correspondences. The approach applies to settings where source and target tasks share a formal objective but differ in layout, state-action space, or physical dynamics, as demonstrated across our discrete and continuous control experiments.

The primary gap between this work and physical deployment is the labeling function: our experiments use handcrafted propositional functions over known state variables, while real systems would require atomic propositions derived from raw sensor observations. Recent work on learned temporal logic monitors and neural proposition detectors provides viable directions for closing this gap. Using formal task specifications as a transfer medium also offers an interpretability benefit: the automaton makes explicit what task-structure knowledge is being transferred and how it influences the student. We do not claim or evaluate safety properties in this work.

### Acknowledgments

This work was supported by DARPA under Agreement No. HR0011-24-9-0427 and NSF under Award CCF-2106339.

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

## Appendix

## A   Algorithm 1

In Algorithm 1 details our proposed automaton distillation method.

---

**Algorithm 1:** Automaton Distillation with DQN or TD3

---

**Input**: Teacher automaton Q-values $Q_{\text{teacher}}$, automaton state transition function, $\delta$, labeling function $L$, number of training steps $T$, number of steps per target update, $t_{update}$, batch size $M$, actor update period $d$, learning rate $\alpha$, annealing rate $\rho$, soft-update,$\tau$, exploration constant $\epsilon$, discount factor $\gamma$

**Output**: Student parameters $\theta$

Initialize Networks:

**if** *DQN* **then**
 | Critic: $Q_\theta$; Target: $\theta_{target} \leftarrow \theta$
**end**
**else if** *TD3* **then**
 | Critics : $Q_{\theta_1}, Q_{\theta_2}$; Actor: $\pi_\phi$;
 | Targets: $\theta_1^{target} \leftarrow \theta_1$, $\theta_2^{target} \leftarrow \theta_2$, $\phi' \leftarrow \phi$
**end**

Initialize replay buffer $ER$, automaton transition visit counts $\eta$

**for** $t \leftarrow 1$ **to** $T$ **do**
 **if** *DQN* **then**
  | Take $\epsilon$-greedy action $a$;
 **end**
 **else if** *TD3* **then**
  | Select action with exploration noise $a \leftarrow \pi_\phi(s) + \varepsilon, \varepsilon \sim \mathcal{N}(0, \sigma_{noise})$
 **end**
 Observe reward $r$ and new state $s'$;
 Compute new automaton state $\omega' = \delta(\omega, L(s'))$;
 Append augmented experience $((s, \omega), a, r, (s', \omega'))$ to the replay buffer $ER$ with priority 1;
 Sample $M$ transitions, $\{(s_i, \omega_i), a_i, r_i, (s'_i, \omega'_i)\}_{i=1}^M$, from replay with priority $p_i$;
 Compute annealing parameters for each transition $\beta_i \leftarrow \rho^{\eta(\omega_i, L(s'_i))}$;
 Update $Q_{\text{target}}$ according to Eq. equation 5 or Eq. equation 6;
 Generate adjusted targets $Q'_i \leftarrow \beta_i Q_{\text{teacher}}(\omega_i, L(s'_i)) + (1 - \beta_i)Q_{\text{target}}$;
 Update Q-networks $\theta \leftarrow \theta - \frac{\alpha}{M}\sum_i p_i \nabla_\theta(Q'_i - Q((s_i, \omega_i), a_i; \theta))^2$;
 **if** *TD3* and $t \bmod d = 0$ **then**
  | $\phi \leftarrow \phi - \frac{\alpha}{M}\sum_i \nabla_\phi Q_1((s_i, \omega_i), \pi_\phi(s_i))$;
 **end**
 Update buffer priorities $p_i = (Q'_i - Q((s_i, \omega_i), a_i; \theta))^2$;
 **for** $i \leftarrow 1$ **to** $M$ **do**
  | Update visit count $\eta(\omega_i, L(s'_i)) \leftarrow \eta(\omega_i, L(s'_i)) + 1$;
 **end**
 **if** $t \bmod t_{update} = 0$ **then**
  **if** *DQN* **then**
   | $\theta_{target} \leftarrow \theta$;
  **end**
  **else if** *TD3* **then**
   | $\theta_k^{target} \leftarrow \tau\theta_k + (1 - \tau)\theta_k^{target}$   $k = 1, 2$;
   | $\phi' \leftarrow \tau\phi + (1 - \tau)\phi'$;
  **end**
 **end**
**end**

---

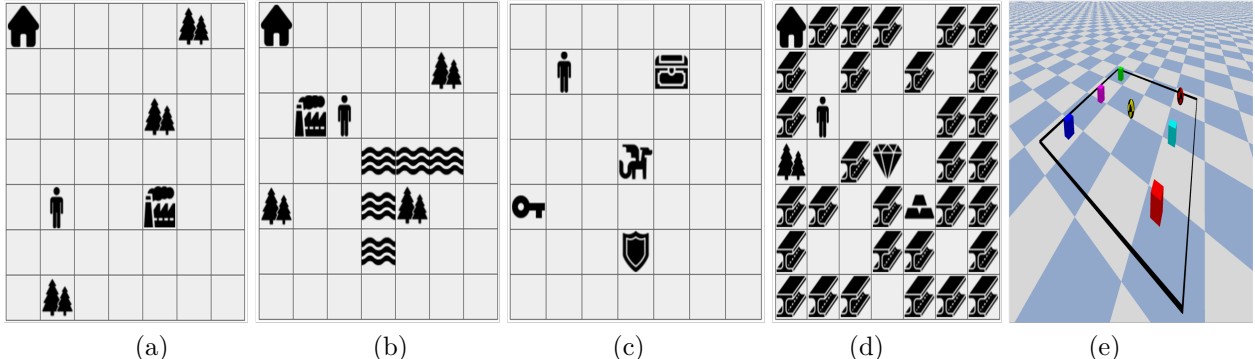

Figure 7: Example $7 \times 7$ environment configurations for the *Blind Craftsman* (a), *Blind Craftsman with Obstacles* (b), *Dungeon Quest* (c), *Diamond Mine* (d), and *Continuous Sample* (e) environments.

## B  Experimental Details

### B.1  Environments

**BlindCraftsman.** This environment consists of woods, a factory, and a home; obstacles can be added for extra difficulty. The objective is satisfied when the agent has crafted three tools and arrived home. One wood is required to craft a tool. However, since the agent can only carry two pieces of wood at a time, the agent must alternate between collecting wood and crafting tools. See Figure 7a,b for an illustration of the *BlindCraft* environment and a variant with obstacles.

**DungeonQuest.** This environment consists of a key, a chest, a shield, and a dragon. The agent can acquire a key and a shield by interacting with a key or shield, respectively. Additionally, the agent can obtain a sword by interacting with a chest with a key in its inventory. Once the agent has the sword and the shield, it may interact with the dragon to defeat it and complete the objective. A visualisation of the this environment is provided in Figure 7c.

**Goldmine.** *Gold Mine*: This environment consists of a wood tile, a diamond tile, GoldMine tiles, and iron tiles. The agent may acquire wood, iron, or GoldMine by interacting with the respective objects. Once the agent has collected wood and 30 iron, it automatically crafts a pickaxe. The agent may then obtain diamond by interacting with the diamond while holding a pickaxe. Once the agent has acquired either 1 diamond or 10 GoldMine, it may return to home to complete the objective. To simplify the resulting automaton and limit unnecessary reward, once the agent has collected GoldMine, it cannot obtain the diamond, and vice versa. Intuitively, although collecting diamond requires less automaton transitions, collecting 30 iron is relatively time-consuming. Thus, this environment represents a scenario where the objective automaton misrepresents the difficulty of the task. Figure 7d shows a visualization of the *GoldMine* environment

In the aforementioned environmemnts, the source domains are randomly generated maps $7 \times 7$ as in Figure 7, and the target domains, either an independently generated $10 \times 10$ map or an environment with *width* and *height* within a continuous range of $[7\,\mathrm{m}, 7\,\mathrm{m}]$ as shown in fig. 7(e). Their corresponding objective automaton is shown in Figure 8. For the grid-world, agents can obtain objects by being on the object tile. On the other hand, objects are collected in a continuous environment when their Euclidean distance to the object is $< 0.25\,\mathrm{m}$. The agent receives a reward of $+1$ for collecting each item, $+100$ for expecting the final task, and a $-0.1$ per time step, an additional $-0.1$ for going out of the boundary in the continuous environment.

**FlatWorld.** We adapt the *FlatWorld* environment introduced by (Voloshin et al., 2023). The environment consists of a two-dimensional continuous world with a discrete action space and colored circular regions representing atomic propositions. Importantly, these regions overlap in various places, which means that multiple propositions can hold true at the same time. The initial agent position is sampled randomly from the space in which no propositions are true. At each time step, the agent can move in one of the 8 compass directions. If it leaves the boundary of the world, the agent receives a penalty and the episode is terminated

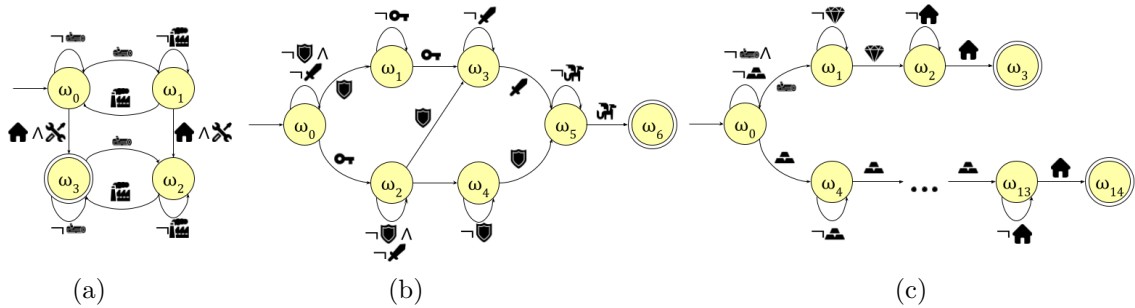

Figure 8: Objective automata for the *Blind Craftsman* (a), *Dungeon Quest* (b), and *Diamond Mine* environments.

prematurely. shows a visualisation of the *FlatWorld* environment. The spatial shift of circles tests whether the agent can generalize the behavioral pattern (visit region types in sequence) across different spatial configurations. The addition of a cross wall introduces spatial reasoning. The agent must discover detour paths around the obstacle while maintaining the sequential visitation strategy learned from the teacher.

### B.2 Task Specifications

Table 2 list the task objectives defined over the APs in each environment used for our evaluation.

### B.3 Hyperparameters

**Neural network architectures.** To learn the policy in the discrete environment, the network consists of a convolutional feature extractor and separate value and advantage heads. The feature extractor is a residual network with 3 residual blocks, each using a $3 \times 3$ convolutional kernel with 32 filters and Leaky ReLU activation. The resulting feature map is flattened and split into equal halves, which are fed separately to the value and advantage heads. Each head contains a single fully connected layer with 1 and $\#$ *actions* nodes, respectively. Q-values are reconstructed by re-centering advantages to have a mean equal to the output of the value head. The neural network takes as input a stack of 2D grids. Each layer in this stack represents a distinct entity type, which includes the agent, tile types, or inventory item types. Inventory items are represented through constant-valued input planes.

In the continuous environment, the network consists of a two-layer feedforward neural network of $[256, 256]$ hidden nodes, respectively, with rectified linear units (ReLU) between each layer for both the actor and critic and a final tanh unit following the output of the actor. The network receives a compact vector comprising the 2D position of the agent, velocity components, and current inventory counts. This vector is generated by the underlying Box2D physics engine that simulates realistic environmental dynamics.

**TD3 and Dueling DQN hyperparameters** The hyperparameter for TD3 (Fujimoto et al., 2018) and Dueling DQN (Wang et al., 2016b) are listed in Table 3.

### B.4 Evaluation metrics

We define the training history under policy $\pi$ as $h_\pi(t)$, representing the expected return at timestep $t$. Using this history, we compute the following transfer evaluation metrics:

$$\text{Time to Threshold (TT)} = \min\{t \mid h_\pi(t) \geq \kappa\}$$

$$\text{Transfer Ratio (TR)} = \frac{AUC\big(h_{\pi_{\text{student}}}\big) - AUC\big(h_{\pi_{\text{target}}}\big)}{\big|AUC\big(h_{\pi_{\text{target}}}\big)\big|} \tag{11}$$

Table 2: Task objectives defined over the APs in each environment used for our evaluation.

| | |
|---|---|
| DungeonQuest | $\varphi_1$ F(dragon) $\wedge$ (key R$\neg$ sword) $\wedge$ (sword R$\neg$ dragon) $\wedge$ (shield R$\neg$ dragon) |
| BlindCraftsman | $\varphi_2$ G(wood $\implies$ F factory) $\wedge$ F (tools $\geq 3 \ \wedge$ home) |
| GoldMine | $\varphi_3$ F(home) $\wedge$ ($\neg$ F(GoldMine $= 1$) $\vee \neg$ F(wood)) |
| | $\wedge$ (wood R $\neg$ diamond) $\wedge$ (GoldMine $= 1$ R $\neg$ GoldMine $= 2$) |
| | $\wedge \cdots \wedge$ (GoldMine $= 9$ R $\neg$ GoldMine $= 10$) |
| | $\wedge$ ((diamond $\vee$ GoldMine $= 10$) R $\neg$ home) |
| FlatWorld | $\varphi_4$ F((red $\wedge$ blue)) |
| | $\varphi_5$ F(red $\wedge$ (blue $\wedge$ F(green))) |

Table 3: Hyperparameters used for TD3 and Dueling DQN training.

| Category | Parameter | TD3 | Dueling DQN | Description |
|---|---|---|---|---|
| **Shared** | Actor Learning Rate | 1e-3 | 1e-4 | Adam learning rate for policy/actor |
| | Critic Learning Rate | 1e-3 | 1e-4 | Adam learning rate for Q-network/critic |
| | Discount Factor ($\gamma$) | 0.99 | 0.99 | Temporal discount for future rewards |
| | Soft Update ($\tau$) | 0.005 | 0.005 | Target network soft update rate |
| | Batch Size | 100 | 64 | Samples per training step |
| | Buffer Size | $10^6$ | 150,000 | Maximum replay buffer capacity |
| **TD3** | Policy Noise | 0.5 | – | Action noise std for target policy smoothing |
| | Noise Clip | 0.5 | – | Clip range for smoothing noise |
| | Policy Delay | 2 | – | Train actor every $N$ critic updates |
| **DQN** | Priority Scale | – | 0.7 | PER importance sampling weight |
| | Min Buffer Size | – | 1,000 | Minimum size before training begins |
| | Epsilon ($\epsilon$) | – | 0.1 | Fixed exploration rate |
| | Target Update Freq. | – | 1,000 | Steps between target network updates |
| | Parallel Envs | – | 8 | CPU cores for rollout collection |
| | Checkpoint Freq. | – | 10,000 | Steps between model checkpoints |

Here, $\kappa$ is a predefined performance threshold, and $t$ is the timestep at which the expected return $h_\pi(t)$ first meets or exceeds $\kappa$. The area under the curve (AUC) is computed as the average of return values along the training curve. In addition to evaluating the Time-to-Threshold (TT) for a single threshold, we define $TT_{AUC}$ as the AUC of the TT values over a set of predefined thresholds. This aggregate metric provides a comprehensive measure of how quickly the policy attains various performance levels.

