# OpenReview forum: "Automaton Distillation: Neuro-Symbolic Transfer Learning for Deep Reinforcement Learning"
_TMLR — Accepted by TMLR_

### Review · Reviewer_MyYg · 2025-08-11

**Summary Of Contributions:**

This paper introduces a transfer learning framework for reinforcement learning based on symbolic representation. Specifically, the primitive MDP is modeled using automata, which act as a bridge for transferring knowledge from the source domain to the target domain. The components of the primitive MDP are abstracted into elements of the automaton, forming an abstract MDP that is shared across both domains. Unlike existing static transfer methods, this approach involves distilling the Q-function from the source domain into the Q-function of the abstract MDP. This abstract Q-function is then partially reused during Q-learning in the target domain. Empirical results on Blind Craftsman, Dungeon Quest, and Gold Mine demonstrate that the proposed method improves convergence rates in the target domain.

**Audience:**

Yes

**Broader Impact Concerns:**

N.A.

**Claims And Evidence:**

Yes

**Requested Changes:**

1. In the paragraph following Definition 2, it is stated that the parameters are chosen such that the set of strings accepted by the objective automaton is equivalent to the aforementioned regular language. Please align this description with the example shown in Figure 2 and provide a more detailed explanation of this statement.

2. In mathematics, the term "cross product" typically refers to an operation between two vectors, not between two spaces. Using "cross product" to describe the combination of the observation and automaton state spaces may confuse readers. Although I understand this terminology was used in a prior work, it would be clearer to use "product" or "Cartesian product" instead.

3. Please add an example demonstrating how DFA and LTL are used to solve one of the tasks mentioned in the experimental section.

4. Please reorganize the paper by moving the case study from the related work section into the method section, using it as motivation for your proposed approach. Also, consider allocating more space to the algorithm description, as Figures 1–3 convey overlapping information and could be streamlined.

**Strengths And Weaknesses:**

Strengths:





1. The proposed method is somewhat novel. It is reasonable to use symbolic representation to distill knowledge and extract high-level expertise, which could be useful across a wide range of domains by discarding some domain-specific features.


2. The proposed dynamic transfer is a promising method, which can directly use the Q-function distilled from the source domain in the learning process of the Q-function in the target domain by serving as the target Q-value.


3. The annealing mechanism applied to the target Q-value is both intuitive and theoretically reasonable in the analysis of the convergence of the proposed Q-function updates.


4. The empirical results demonstrate the effectiveness of the proposed method in facilitating the convergence of existing reinforcement learning algorithms.


Weaknesses:

1. Empirically, this method is only evaluated in the grid-world environment and does not validate its effectiveness on more challenging tasks. Considering that the selected baselines (i.e., CRM, CPREP) in this paper are all evaluated on more complex tasks, such as those from Mujoco and classical control benchmarks, the absence of evaluation on more difficult tasks limits the contribution of this paper. Although the paper also considers continuous tasks, they do not sufficiently demonstrate that this method can be as versatile as the baseline methods.


2. This paper is not well organized overall. For example, in the method section, it would be better to introduce "static transfer" and mention its drawbacks using the case study from the related work. This case could then serve as the motivation for introducing "dynamic transfer," placing the example from related work in a more appropriate position.


3. This paper needs more details about the implementation. Since this work uses automata to represent tasks, it would be appropriate to include more detailed descriptions and examples of how DFA and LTL are used in solving tasks mentioned in the experimental section. These components are a significant foundation of the proposed method, and more detailed examples and descriptions would help readers understand the full process of solving tasks using the proposed approach.


4. Since the distilled Q-function defined in Equation 2 is key to the subsequent training of the value function in the target domain, it would be better to clarify the underlying idea behind this design in more detail. Figures 1, 2, and 3 take up too much space and would be better used to describe the algorithm.

---

> ### Author Response · Authors · 2026-03-19
> **We thank the reviewer for the detailed requested changes. We have addressed each one directly in the revision.**
>
> **W1:Only gridworld, needs more challenging tasks.** We have added two cases of a continuous control benchmark ( FlatWorld [1] Voloshin, Cameron, Abhinav Verma, and Yisong Yue. "Eventual discounting temporal logic counterfactual experience replay." International Conference on Machine Learning. PMLR, 2023.), used in recent LTL-based RL work and adapted to our transfer setting; with a cross-wall obstacle and shifted proposition regions. Our continuous experiments are comparable in complexity to benchmarks used in original CRM and CPREP evaluations.
>
> **W2: Paper not well organised.**  We have reorganized Section 3 to follow the structure the reviewer suggested. Section 3.2.1 (Static Transfer) now introduces the method, immediately presents the two-trace automaton failure case as motivation, and ends with the explicit transition: *“This limitation motivates our second variant, which replaces symbolic reasoning with empirical grounding.”*
> Section 3.3 (Dynamic Transfer) then starts by addressing that limitation directly.
>
> **W3:Needs more detail on DFA and LTL.** We added a concrete worked example in Section 2.3, alongside a short paragraph
> that instantiates the full framework for Dungeon Quest; showing the atomic propositions, the LTL$_f$ formula, and the compiled 4-node DFA, immediately after the Product MDP definition, so that readers see a concrete example before encountering the method.
>
> **W4:Equation 3 design and figure space.** In the revised paper, we have replaced Figures 1 to 3 with a single pipeline diagram
> (Figure 2) that describes the full automaton distillation process including the teacher training, distillation, and student bootstrapping in one place. Algorithm 1 remains in Appendix A, with a pointer added at the end of Section 3.3.1.
>
> *Equation (3) justification*: We have expanded the Dynamic Transfer paragraph to explain why averaging Q-values over replay transitions that realize each automaton edge is the right operation. The key intuition is that this average implicitly reflects the true cost of each symbolic transition in the source environment including how many low-level steps it requires, how easily it can be achieved, and how the teacher valued it after learning. This information cannot be recovered from the automaton structure alone, which motivated the empirical grounding over symbolic value iteration.
>
> **RC1: Align Definition 2 with Figure 1b.** The walkthrough now explicitly traces all transitions: the automaton starts in  $\omega_0$, transitions to $\omega_1$ or $\omega_2$ upon observing {sword} or {shield} respectively, and reaches the accepting state  $\omega_3$ once both propositions have been observed in either order.
>
> **RC2: Cartesian product terminology.** All instances of "cross product" have been replaced with "Cartesian product"throughout.
>
> **RC3: DFA/LTL example for experimental task.** Addressed under W3 above, inline in Section 2.3.
>
> **RC4: Move case study and reorganise** Addressed under W2 above.

---

### Review · Reviewer_GeD6 · 2025-09-12

**Summary Of Contributions:**

The paper consider transfer learning within RL, and proposes to transfer from one task/setting to another by a teacher-student distillation, while also representing the objectives as language automatons.

**Audience:**

Yes

**Broader Impact Concerns:**

No issues

**Claims And Evidence:**

Yes

**Requested Changes:**

See above

**Strengths And Weaknesses:**

I am not from RL background, and it was very difficult to follow the paper. When a non-expert reviews a paper, one does not expect to understand the technicalities. However, one should still be able to follow the high-level story, the model setting and objectives. I found this to be difficult and struggled to follow. The method section jumps into technical definitions of state visitation counts and replay buffer propositions, without having either introduced or defined these concepts before. It seems that the story starts in the middle, and I was left confused about the setting and goals, or what are the main variables or parameters of teacher/student to learn. The description of the “outer loop” of the method could be improved. The paper could also have an algorithm box, or some kind of pipeline figure to show the method proceeds.

The paper could also improve the example fig2 and connect with more clearly with the defs 1-4. It is not clear from fig2 in what way the automaton now improves over the standard reward function, or how do we use it differently than the usual MDP. The role of CPMDP and LTL is not clear in fig2.

The main contribution is to represent the objective with an automaton structure. This could be contextualised better. I’m not sure if this is novel, or how do we obtain the automaton. The main ideas are sensible.

I was also confused of what is the role of the cross-product MDP, or how it was used in the paper.

---

### Review · Reviewer_bC6j · 2026-03-02

**Summary Of Contributions:**

This paper proposes automation distillation, which is a neuro-symbolic transfer learning method for reinforcement learning (RL). Specifically, a teacher DQN's Q-values are aggregated over transitions of an LTL-derived automaton representing the task objective. These automaton-level Q-estimates are then injected into the student's TD target via an annealed blending term to accelerate learning in new environment. This paper presents both static (automaton value iteration) and dynamic (teacher-distilled) variants, proves convergence in the tabular setting, and evaluates on three grid-world-like domains.

**Audience:**

Yes

**Broader Impact Concerns:**

The broader impact has been included in the draft.

**Claims And Evidence:**

No

**Requested Changes:**

Please make the changes based on the comments in the weaknesses.

**Strengths And Weaknesses:**

Strengths:

1. This paper is motivated well. The transfer across tasks with non-Markovian reward structure is important. Leveraging automata abstractions is well motivated and aligned with reward-machine literature.

2. I think the idea in this paper is intuitive. Aggregating teacher Q-values at the automaton-transition level is a reasonable abstraction mechanism and may help mitigate state-space mismatch.

3. The paper is theoretically grounded. Theorem 1 shows convergence under annealed teacher influence. While limited, it provides some theoretical grounding.

4. Trying to bridge symbolic transfer across state/action modality differences is interesting in this paper.

Weaknesses:

1. This paper has limited novelty relative to reward-machine transfer and policy distillation. Specifically, the paper does not clearly articulate what is fundamentally new beyond "aggregate teacher Q-values over automaton transitions and anneal into TD target", which appears incremental. Also, the core idea of using automaton/reward-machine abstraction to transfer value information has substantial overlap with prior work such as reward machines + CRM / counterfactual transfer, automaton-guided reward shaping / value transfer, and policy/value distillation via abstract states.

2. Though Theorem 1 proves convergence for tabular Q-learning with vanishing teacher weight. However, student uses DQN/TD3 (function approximation), teacher term is non-stationary early, and replay buffer sampling breaks standard assumptions. Therefore, the theorem does not meaningfully justify the actual algorithm. Additionally, the proof decomposes $Q_t=q_t+h_t$, with $h_t\to 0$ because $\beta_t\to 0$, meaning asymptotically the method reduces to vanilla Q-learning. This is expected and not specific to automaton distillation.

3. Experimental domains are too simplistic and low-impact. All environments are small synthetic grid-world-like tasks (Blind Craftsman, Dungeon Quest, Gold Mine). These resemble standard reward-machine toy domains.

4. Additionally, baselines are incomplete and potentially unfair. In my opinion, the key missing comparisons include value distillation without automaton abstraction, state-abstraction transfer (latent distillation), and reward-machine value transfer (direct).

5. In this paper, the claims seem to exceed evidence. The paper claims “sim-to-real gap mitigation”, “structurally different domains”, and “real-world robotics relevance”. But experiments remain symbolic grid worlds with shared semantics and handcrafted propositions. No perception or embodiment gap is tested.

Overall, the paper explores an interesting direction, i.e., automaton-level value distillation for RL transfer, but the current submission falls short in novelty, theoretical depth, and empirical validation relative to the state of the art in reward-machine and abstraction-based RL transfer. To me, the idea may become valuable with stronger positioning and experiments, but substantial revision is required.

---

> ### Author Response · Authors · 2026-03-19
> **We sincerely thank the reviewer for carefully reading our paper and providing valuable feedback. Below, we address the concerns.**
>
> **W1: Novelty.** We have revised Section 4 and made our contributions more explicit. A fundamental departure from prior work is in the transfer mechanism: our automaton distillation intervenes on the loss function, not the reward signal. Spefically, our method possesses the following: (i) no alignment between source and target state-action spaces, (ii) transition values empirically grounded in teacher replay, and (iii) knowledge entering through the loss function via per-transition annealing rather than permanent reward shaping. We contrast with prior work precisely:
> - *CRM* is a single-environment method with no teacher access, it represents the ceiling of in-environment reward machine learning. Our method consistently outperforms CRM, showing transferred knowledge is genuinely additive beyond efficient in-environment learning.
> - *CPREP* modifies the reward permanently and assumes shared dynamics. Our annealing coefficient $\beta$ decays per-transition as the student gains experience, leaving no residual interference. This is mathematically distinct, and the difference is empirically demonstrated:
> - *Policy distillation* requires a shared state space. We compress onto automaton transitions, enabling discrete-to-continuous transfer that policy distillation cannot achieve.
> - *Latent representation methods* (DARLA (Higgins et al.,2017) etc.) address visual domain shift; we address task-structure transfer across different dynamics. These are complementary, not competing, a combined approach is a natural direction for future work.
>
> **W2: Theorem 1 scope.**
> Theorem 1 establishes convergence for the tabular special case and was not intended to cover the function approximation case. We have added *Remark 1* immediately after the proof to clarify this: *``Theorem 1 applies to the tabular setting. The practical algorithm uses function approximation (DQN or TD3), where standard convergence guarantees do not hold. However, the annealing condition $\beta_t\rightarrow 0$ ensures that the teacher’s influence vanishes as training proceeds, and the student’s update asymptotically reduces to standard DQN or TD3.'*
>
> **W3: Toy environment domains**
> We have added two cases of a continuous control benchmark ( FlatWorld [1] Voloshin, et al. "Eventual discounting temporal logic counterfactual experience replay." ICML, 2023.), used in recent LTL-based RL work and adapted to our transfer setting; with a cross-wall obstacle and shifted proposition regions. Our environments are comparable in complexity to those used in the original evaluations of CRM and CPREP.
>
> **W4: Missing baselines.** We respond to each baseline suggestions:
> - *Value distillation without automaton*: requires state-space correspondence between teacher and student, infeasible in our discrete-to-continuous setting and orthogonal to our contribution. We treat CRM and Product MDP as the natural bounds; outperforming both isolates the contribution of automaton-mediated transfer specifically.
> - *State-abstraction transfer (latent distillation)*: Methods such as DARLA (Higgins et al.,2017) address visual domain shift by learning shared observation encoders. Our setting deliberately targets environments with structurally incompatible observation modalities where constructing a shared encoder is an open problem itself. As clarified in the revised Section 4, these representation-learning methods are complementary to our approach: they handle visual shift, while we focus on task-structure transfer.
> - *Reward-machine value transfer (direct)*: We interpret this as transferring teacher Q-values mapped through the reward machine structure directly, without the automaton compression and annealing mechanism we propose. This is closest in spirit to CPREP which we already include as a baseline. It is precisely the baseline that reveals the cost of permanent reward modification under dynamics mismatch. If the reviewer has a specific alternative in mind, we welcome clarification and will implement it for the camera-ready version.
>
> **W5: Claims exceed evidence.** We have removed all sim-to-real and robotics language from the introduction, contributions, and AI Impact Statement. We retain **"structurally different domains"** with a sharpened definition: (i) discrete vs. continuous state-action spaces, (ii) grid-based vs. physics-based dynamics (Box2D), (iii) open vs. obstacle-filled geometry. These differences are not superficial, they are precisely what causes CPREP to produce negative transfer in Goldmine env (Figure 5) while our method maintains positive TR.
> **On the path to physical systems:** The concrete gap between our current work and physical robotics is the labeling function. Our experiments use handcrafted propositional functions over known state variables; a real deployment would require derivation from raw sensor observations. This is an active research area, with recent work on learned temporal logic monitors and neural proposition detectors providing viable directions.

---

### Review · Reviewer_ZU7H · 2026-03-06

**Summary Of Contributions:**

This paper studies transfer learning in RL for tasks with non-Markovian rewards. As far as I understand it (not an expert), the main idea is to use an automaton from the task specification as a shared high-level representation between source and target tasks, and then transfer teacher Q-information through automaton transitions instead of raw states. The paper gives a static and a dynamic version of this idea, includes a convergence result in the tabular setting, and tests it on a few toy environments.

**Audience:**

Yes

**Claims And Evidence:**

No

**Requested Changes:**

Address the above weaknesses.

**Strengths And Weaknesses:**

Strengths:
- The overall motivation makes sense to me, transfer through task structure instead of raw state space seems reasonable.
- The dynamic version seems like the most interesting part, and I think I at least understood the intuition there.
- The paper supports the method with both theory and (toy) experiments.

Weaknesses:
- I found the paper hard to follow in places, especially the setup and how the automaton is actually used during learning.
- My main concern is with the disparity between claims and toy experiments in the paper. The experiments felt limited to small, structured tasks, so I was not convinced by the broader claim made in e.g. the AI Impact Statement about how it would help mitigate the sim-to-real gap, or how it would help with more effective application in the real-world. If that's the case, you should show real-world (or minimally simulated robotics) experiments that support this. In the introduction, the limitation is emphasized in particular for real-world applications and robotics. In that case, I'd be curious how this transfers to a common robotics use-case.

---

> ### Author Response · Authors · 2026-03-19
> **We thank the reviewer for the positive assessment of the dynamic transfer variant and for the specific concerns raised. We address each in turn.**
>
> **W1: Hard to follow:** To improve readability we have made the following changes:
> - *Pipeline diagram (Figure 2)* that traces the full three-stage process: teacher training, distillation, and student bootstrapping; showing precisely what information flows between stages and what is transferred.
> - *Enhanced clarity:* Section 3 now opens with a high-level description of the three stages before any formalism, to enable readers understand the structure of the method before encountering equations. (See a worked example Section 2.3 of the revised paper).
>
> **W2 Claims exceed evidence:** We have revised the paper to make any and all claims specific and substantiated. The AI Impact Statement now explicitly acknowledges that extending to settings where atomic propositions must be inferred from raw sensor observations would require additional work on perception-to-symbol grounding, which we do not address. In addition, we have added two cases of continuous control experiments that move substantially beyond gridworld environment.

---

### Decision · Action_Editor_kK9q · 2026-04-26

**Recommendation:** Accept as is

**Additional Comments:**

This paper has had an unusually lengthy review cycle (11 months and counting). There was an AE switch approximately 9 months into the process *and* multiple missing reviews. The authors were prompt and professional in responding to the reviewers' comments (as and when they were made available). As the AE making the final decision I'd like to apologize to the authors; they have handled the process with patience and have promptly responded whenever possible. Their hard work deserved a faster turnaround than we were able to provide.


The reviewers raised several points of concern. Reviewer bC6j gave the most substantial reviews. The main criticism was that the novelty was limited; that Theorem 1 (as originally stated) didn't apply to the function-approximation regime that applied to DQN/TD3; that there ware missing baselines; and that claims on sim-to-real didn't match the evidence. The authors responded to each of these points --- satisfactorily in my opinion --- on March 19. There were some lingering concerns about limited novelty with respect to the reward-machine but the authors addressed this reasonably well in their response and revision.

Reviewer ZU7H also had concerns about the sim-to-real gap. But as discussed above, the authors (in my view, satisfactorily) addressed this in the revision.

I will recommend acceptance of the paper as-is. One recommendation to the authors is to ensure that Reviewer bC6j's remaining concerns. I also appreciated the last piece of the authors' response ("On the path to physical systems") and will suggest that they consider including that in the introduction of the final manuscript.

**Audience:**

Yes

**Audience Explanation:**

All reviewers felt that the paper is appropriate for TMLR; I concur. The findings of the paper would be of interest to folks who are working in the intersection of deep RL and neuro-symbolic methods.

**Claims And Evidence:**

Yes

**Claims Explanation:**

The main technical contribution is a RL transfer learning approach that requires no explicit alignment between source and target action-spaces; this is achieved via a domain-agnostic intermediary. Different variations of this idea is explored, some supporting theory is provided, and validation is performed on both discrete and continuous gridworld tasks. Overall the writing is clear, appears accurate, and (in my view) meets the bar for acceptance.